# Risk Factors for Periprosthetic Joint Infection after Primary Total Knee Arthroplasty

**DOI:** 10.3390/jcm11206128

**Published:** 2022-10-18

**Authors:** Emerito Carlos Rodriguez-Merchan, Alberto D. Delgado-Martinez

**Affiliations:** 1Department of Orthopedic Surgery, La Paz University Hospital, Paseo de la Castellana 261, 28046 Madrid, Spain; 2Osteoarticular Surgery Research, Hospital La Paz Institute for Health Research—IdiPAZ (La Paz University Hospital—Autonomous University of Madrid), 28046 Madrid, Spain; 3Department of Orthopedic Surgery, Hospital Universitario de Jaen, 23007 Jaen, Spain; 4Department of Surgery, University of Jaen, 23071 Jaen, Spain

**Keywords:** periprosthetic joint infection, risk factors, total knee arthroplasty

## Abstract

Periprosthetic joint infection (PJI) is a major adverse event of primary total knee arthroplasty (TKA) from the patient’s perspective, and it is also costly for health care systems. In 2010, the reported incidence of PJI in the first 2 years after TKA was 1.55%, with an incidence of 0.46% between the second and tenth year. In 2022, it has been published that 1.41% of individuals require revision TKA for PJI. The following risk factors have been related to an increased risk of PJI: male sex, younger age, type II diabetes, obesity class II, hypertension, hypoalbuminemia, preoperative nutritional status as indicated by prognostic nutritional index (PNI) and body mass index, rheumatoid arthritis, post-traumatic osteoarthritis, intra-articular injections prior to TKA, previous multi-ligament knee surgery, previous steroid therapy, current tobacco use, procedure type (bilateral), length of stay over 35 days, patellar resurfacing, prolonged operative time, use of blood transfusions, higher glucose variability in the postoperative phase, and discharge to convalescent care. Other reported independent risk factors for PJI (in diminishing order of importance) are congestive heart failure, chronic pulmonary illness, preoperative anemia, depression, renal illness, pulmonary circulation disorders, psychoses, metastatic tumor, peripheral vascular illness, and valvular illness. Preoperative intravenous tranexamic acid has been reported to diminish the risk of delayed PJI. Knowing the risk factors for PJI after TKA, especially those that are avoidable or controllable, is critical to minimizing (ideally preventing) this complication. These risk factors are outlined in this article.

## 1. Introduction

According to Carulli et al., total knee arthroplasty (TKA) is one of the most successful surgical techniques in orthopedic surgery, with good clinical outcomes and a high survival percentage of more than 90% of cases at long-term follow-up. The increasing mean population age, worsening of joint degenerative disorders, and joint sequelae related to previous fractures have caused a continuous rise in the number of TKAs in every country annually, along with an expected increase in adverse events [1]. A frequent cause of revision TKA following primary TKA is periprosthetic joint infection (PJI) [2,3,4,5,6]. PJI was published by the Musculoskeletal Infection Society (MSIS) in 2011 [7] (Table 1).

In 2018, Parvizi et al. reported an evidence-based definition for knee PJI that has demonstrated very good performance on formal external validation [8]. Two positive cultures or the existence of a sinus tract were deemed primary factors and diagnostic of PJI. The estimated weights of increased serum C-reactive protein (CRP) (>1 mg/dL), D-dimer (>860 ng/mL), and erythrocyte sedimentation rate (ESR) (>30 mm/h) were 2, 2, and 1 point, respectively. Moreover, increased synovial fluid white blood cell count (>3000 cells/μL), alpha-defensin (signal-to-cutoff ratio > 1), leukocyte esterase (++), polymorphonuclear percentage (>80%), and synovial CRP (>6.9 mg/L) were given 3, 3, 3, 2, and 1 point, respectively. Individuals with a total score of greater than or equal to 6 were deemed infected, whereas a score between 2 and 5 needed the addition of intraoperative findings for proving or disproving the diagnosis. Intraoperative findings of positive histology, purulence, and single positive culture were given 3, 3, and 2 points, respectively. Put together with the preoperative score, an aggregate of greater than or equal to 6 was deemed infected, a score between 4 and 5 was uncertain, and a score of 3 or less was not infected. These standards showed a greater sensitivity of 97.7% compared with the MSIS (79.3%) and the International Consensus Meeting definition (86.9%), with an akin specificity of 99.5% [8].

In 2021, McNally et al. reported the result of a plan created by the European Bone and Joint Infection Society (EBJIS) and endorsed by the MSIS and the European Society of Clinical Microbiology and Infectious Diseases Study Group for Implant-Associated Infections (ESGIAI). McNally et al. defined PJI using a three-degree method to the diagnostic sequence, leading to a definition set and guidance that was fully backed by the EBJIS, MSIS, and ESGIAI [9]. There are three possibilities: infection unlikely, infection likely, and infection established based on the following data: clinical and blood workup (clinical features, CRP); synovial fluid cytological analysis (leukocyte count (cells/µL); polymorphonuclear percentage); synovial fluid biomarkers (alpha-defensin); microbiology (aspiration fluid, intraoperative fluid and tissue, sonication (CFU/mL); and histology (high-power field, 400× magnification). This new EBJIS definition can now be used worldwide [9].

PJI is a severe complication of primary TKA from the patient’s perspective, and it is also very costly for health care systems [10]. In fact, PJI is one of the most overwhelming adverse events of TKA [11].

Although one-stage revision TKA is performed in certain situations and centers, a PJI usually requires a two-stage revision TKA, which involves a double surgical intervention. First, the removal of the infected implant (septic loosening) is required. Following this procedure, a period of several weeks of antibiotic treatment is needed until the infection is considered cured (normalization of the ESR and CRP and healing of the surgical wound). The second intervention is the insertion of a new implant, using a model that is stable for proper functioning of the knee [11]. Figure 1 shows a case of PJI (septic loosening) that was solved by a two-stage revision TKA.

It is important to emphasize that debridement, antibiotics, and implant retention (DAIR) is today a frequently utilized procedure in early infections [12,13,14,15,16]. Toh et al. reported that DAIR is the procedure of preference for individuals with acute postoperative and acute hematogenous PJI [15]. They stated that DAIR failure was related to premature mortality. Repeated DAIRs, increased ESR > 107.5, and *S. aureus* PJI were related to treatment failure, and two-stage revision TKA was advised. It is also relevant to remark that the likelihood of PJI after primary TKA can be reduced by decreasing the patient’s weight, which will also minimize the risk of implant failure [17].

The aim of this narrative review is to present an overview of the risk factors for PJI after primary TKA. To this end, we have outlined the most important points to facilitate further investigation into specific aspects of the topic. This article seeks to explain the risk factors for PJI after primary TKA, with the aim of controlling or preventing this complication whenever possible.

A PubMed (MEDLINE), Cochrane Library, Web of Science, and Scopus search of reports on PJI in TKA was conducted. The key words utilized were “PJI TKA risk factors”. The inclusion criterion was reports focused on the risk factors for PJI in TKA. Studies not focused on such risk factors were disregarded. The searches were dated from the creation of the search engines until 30 September 2022. From the 13,304 articles (10,300 in the Web of Science, 2860 in Scopus, 136 in PubMed, 8 in The Cochrane Library), we chose those that seemed most directly related to the title of this article (66 articles)**.**

## 2. Incidence of PJI after TKA

Several authors have reported infection rates of 2–5% after TKA [4,5,6,7,18,19]. However, in a level 2 evidence study (prognostic study) published in 2010, among 69,663 patients operated on for TKA, Kurtz et al. identified 1400 infections. The incidence of infection in the first 2 years was 1.55%. The incidence between the second and tenth year was 0.46%. PJI was observed to occur at a fairly elevated percentage in Medicare individuals, with the highest risk in the first 2 years following TKA; roughly a quarter of PJIs occur after 2 years [20].

In 2022, the McMaster Arthroplasty Collaborative (MAC) found that 1.41% of individuals experienced revision TKA for PJI [11]. Figure 2 shows a comparison between the rates of PJI after TKA reported in 2010 and 2022.

## 3. Risk Factors for Periprosthetic Joint Infection following Primary TKA

### 3.1. Patient-Related Risk Factors

#### 3.1.1. Male Gender, Procedure Type (bilateral), Length of Stay over 35 Days, and Usage of Transfusions Have Been Shown to Be Risk Factors for Postoperative PJI

In 2021, Ko et al. found that male sex, low family earnings, surgical technique type (bilateral), length of stay (LOS) ≥ 35 days, and transfusions were risk factors for postoperative adverse events after TKA in individuals with idiopathic knee osteoarthritis. The aforementioned authors analyzed 560,954 individuals older than 50 years. The risk of PJI was evaluated with 8 independent parameters: sex, age, place of residence, family earnings, hospital bed size, type of surgical technique (unilateral or bilateral, primary or revision TKA), LOS, and the use of transfusions [21].

#### 3.1.2. Male Gender, Younger Age, Type II Diabetes, Posttraumatic Osteoarthritis, Patellar Resurfacing, and Discharge to Nursing Home Were Related to an Increased Risk of PJI

In 2022, the MAC carried a population-based cohort investigation utilizing linked administrative databases. The multivariable analysis showed that male gender, younger age, type II diabetes, posttraumatic osteoarthritis, patellar resurfacing, and discharge to convalescent care were related to an increased risk of PJI [11].

In an article with level 2 evidence (prognostic study), the independent risk factors for PJI (in diminishing order of importance) were congestive heart failure, chronic pulmonary illness, preoperative anemia, diabetes, depression, renal illness, pulmonary circulation disorders, obesity, rheumatologic illness, psychoses, metastatic tumor, peripheral vascular illness, and valvular illness [22].

In 2013, Chen et al. showed that the principal factors associated with PJI following TKA were body mass index (BMI), diabetes mellitus, hypertension, steroid treatment, and rheumatoid arthritis. The study had insufficient evidence to demonstrate that the male sex was associated with PJI after TKA. A statistical analysis showed no correlations between urinary tract infection, fixation technique, American Society of Anesthesiology (ASA) score, bilateral procedure, age, transfusion, antibiotics, bone graft, and PJI [23].

A study with level 2 evidence (prognostic study) reported by Kurtz et al. in 2010 showed that women had a lower risk of PJI than men. Comorbidities also increased TKA infection risk. Individuals receiving public assistance for Medicare premiums were at increased risk for PJI. Hospital factors did not contribute to an increased risk of infection. PJI occurred at a fairly elevated percentage in Medicare individuals, with the greatest risk of PJI within the first 2 years following TKA; nonetheless, around 25% of all PJIs occurred after 2 years [20].

Cordtz et al. observed that individuals with rheumatoid arthritis had a diminished 10-year risk of revision TKA, whereas the risk of PJI was increased compared with individuals with osteoarthritis after TKA. Previous treatment with biological disease-modifying antirheumatic drugs was not related to an increased risk of PJI [24].

In an investigation with level 3 evidence (therapeutic study), Pancio et al. found that a higher proportion of individuals who had undergone multi-ligament knee surgery experienced infections compared with matched controls (7% vs. 1%), respectively. Previous multi-ligament surgery was associated with a greater risk of PJI [25].

#### 3.1.3. Previous Septic Arthritis Has Been Shown to Be a Risk Factor for Postoperative PJI

Previous septic arthritis has also been recognized as a PJI risk factor [25]. Pooled data from more than 1300 arthroplasties in published papers revealed a PJI rate of 5.96% when a previous infection occurred in the same articulation. The risk of infection was lower because the TKA surgery was delayed from the resolution of the previous infection.

#### 3.1.4. Smoking Is related to Higher Percentages of PJI

In 2015, Singh et al. found that smoking was related to an elevated risk of PJI after primary TKA. Tobacco use status was accessible for 7926 (95%) individuals and was not accessible for 446 (5%); 565 (7%) currently smoked tobacco. The hazard ratios for PJI were higher in current tobacco users than in nonusers [26]. Cessation of smoking before TKA is strongly recommended.

#### 3.1.5. Hypoalbuminemia and Obesity Class II Are Dependable Predictors of PJI

According to Man et al., malnutrition is a relevant but changeable risk factor for postoperative adverse events and unfavorable results in orthopedic surgery [27]. They sought to detect biomarkers of malnutrition in individuals undergoing TKA that could be predictive of adverse postoperative complications in the hospital, to identify patients at risk and optimize their nutritional status prior to TKA. These authors analyzed 624 patients in whom possible biomarkers of pre-operative malnutrition, including hypoalbuminemia (serum albumin < 3.5 g/dL), total lymphocyte count (TLC) (<1500 cells/mm^3^), and BMI, were evaluated for associations with in-hospital postoperative adverse events. The frequencies of hypoalbuminemia, low TLC, overweight, obesity class I, and obesity class II were 2.72%, 33.4%, 14.8%, 44.5%, and 26.9%, respectively. There were significant relationships between hypoalbuminemia and type II obesity (BMI ≥ 30.0 kg/m^2^) and PJI percentages and no significant relationships between these adverse events and low TLC, overweight, or type I obesity. It was also found that individuals with hypoalbuminemia or type II obesity with gouty arthritis were more prone to experience PJI. The authors concluded that hypoalbuminemia and type II obesity together were dependable biomarkers of preoperative malnutrition that could predict PJI following TKA; however, low TLC, overweight, and type I obesity were not significantly related to an increased risk of PJI [27].

#### 3.1.6. Intra-Articular Injections Prior to TKA Are Related to a Higher Risk of PJI

In a level 3 evidence study (therapeutic study) published in 2017, Bedard et al. observed that intra-articular knee injections with corticosteroids, hyaluronic acid, or other drugs prior to TKA were related to an increased risk of PJI, and this association appeared to be time dependent: the shorter the delay between injection and TKA, the greater the likelihood of PJI [28]. The proportion of patients undergoing TKAs who developed PJI was greater in those who were given an injection prior to TKA than in those who were not (4.4% vs. 3.6%). Similarly, the proportion of patients undergoing TKAs who developed PJI requiring surgical reintervention was also greater among those who received an injection prior to TKA than in those who did not (1.49% vs. 1.04%). An analysis of the months between injection and TKA showed that the odds of PJI were greater for patients injected up to 6 months between injection and TKA, as were the odds of surgical intervention for TKA infection when the injection was within 7 months of TKA. When the time span between injection and TKA was longer than 6–7 months, the ORs were no longer raised [28].

#### 3.1.7. Greater Glucose Variability in the Postoperative Period Is Related to Higher Percentages of PJI

In 2018, Shohat et al. investigated the relationship between glucose variability and postoperative adverse events after TKA (level 4 of evidence study) [29]. They analyzed data on 2698 individuals who had experienced TKA at a single center. Individuals with a minimum of two postoperative glucose values per day or more than three values overall were included in the research. Glucose variability was evaluated utilizing a coefficient of variation. The MSIS criteria were utilized to establish PJI. Some 19.9% of the patients had diabetes. Greater glycemic variability was related to increased LOS, 90-day mortality, PJI, and SSI. Adjusted analyses showed that for every 10-percentage-point rise in the coefficient of variation, the LOS increased by 6.1%, and the risks of PJI and SSI increased by 20% and 14%, respectively. These associations were independent of the year of the surgical procedure, age, BMI, Elixhauser comorbidity index, diagnosis of diabetes, in-hospital utilization of insulin or steroids, or mean glucose values throughout hospitalization. They concluded that greater glucose variability in the postoperative period was related to increased percentages of SSI and PJI after TKA. According to Shohat et al., it is paramount to control glucose variability in the early postoperative phase [29].

#### 3.1.8. Reduction of Patient’s Weight Diminishes the Probability of PJI and Minimizes Implant Failure

It has been published that the likelihood of PJI following primary TKA can be reduced by decreasing the patient’s weight, which will also minimize the risk of implant failure [17].

## 4. Surgical Risk Factors

### 4.1. Prolonged Surgical Time Correlates with Increased Infection Risk

In 2006, Peersman et al. stated that the time span of the surgical procedure had a definite impact on infection rates, especially regarding postoperative infection after TKA. The study confirmed the significance of the time span of TKA as a risk factor for SSI and subsequent PJI. Therefore, the time span of the surgical intervention can be a predictor of PJI [30].

### 4.2. Unilateral versus Bilateral TKA

In 2021, Ko et al. reported that the procedure type (bilateral TKA versus unilateral TKA), was a risk factor for postoperative adverse events following TKA [21]. Some reports have shown that when bilateral TKA is carried out, the LOS, anesthesia duration and rehabilitation period can be reduced and that there are advantages to individuals and hospitals in terms of cost [31,32].

Despite these benefits, there were issues over the safety of bilateral TKA. According to Odum et al., concurrent bilateral TKA had greater percentages of adverse events compared with unilateral TKA [31]. Memtsoudis et al. stated that staging bilateral TKA had either a greater or similar frequency of adverse events compared with simultaneous bilateral TKA [33]. The study by Ko et al. proved the outcomes of those studies, given that the complication hazard ratios for bilateral TKA were consistently higher than those for unilateral TKA [21].

### 4.3. Patellar Resurfacing

A multivariable model reported by the MAC showed that patellar resurfacing was related to an increased risk of PJI. However, patellar resurfacing was the weakest of all significant predictors (*p* = 0.04) [11]. In contrast, a meta-analysis of randomized controlled trials (RCTs) encountered no difference in infection percentages between patellar resurfacing and non-resurfacing [34]. Clearly, future studies should examine the association between patellar resurfacing and PJI after primary TKA.

### 4.4. Risk Factors in the Postoperative Phase

It has been reported that the use of blood transfusions [21], LOS over 35 days [20], higher glucose variability [28], and discharge to convalescent care [11] are important risk factors for PJI following TKA. Ko et al. found that the risk was increased in the longer LOS cohort and in the transfusion cohort [21]. The Cox proportional hazards model reported by the MAC showed that discharge to a nursing home was related to an increased risk of developing PJI after primary TKA [10]. Table 2 summarizes the patient-related and surgical risk factors for PJI after primary TKA as well as risk factors in the postoperative period.

## 5. Other Topics of Interest Related to the Risk of PJI following TKA

### 5.1. Tranexamic Acid Diminishes the Risk of Revision TKA for Acute and Late PJI

In 2020, Lacko et al. analyzed the impact of intravenous usage of tranexamic acid (TXA) on the risk of revision TKA for acute and late PJI following primary TKA [35]. This study included 1529 TKAs (396 men, 1133 women; mean age 67.8 years). Lacko et al. analyzed the revision percentage for acute and late PJI in a cohort of 787 TKAs with pre-operative intravenously used TXA (TXA cohort) compared with a cohort of 742 TKAs without TXA (non-TXA cohort). A multiple logistic regression analysis was conducted to assess significant predictors of TKA revision for acute and late PJI. Revision TKA due to PJI was observed in one individual in the TXA cohort and one individuals in the non-TXA cohort. The cumulative revision percentage of TKA was significantly lower in the TXA group (0.13% vs. 1.08%). A multivariate logistic regression analysis detected 2 predictors of revision TKA: being older than 75 years at the time of primary TKA and male sex. The utilization of TXA was demonstrated to be a significant protective factor. These authors identified a lower cumulative revision percentage of TKA for acute and late PJI when TXA was utilized. Lacko et al. concluded that the pre-operative intravenous utilization of TXA could be an efficacious, safe, and inexpensive approach to preventing PJI [35].

In 2021, Hong et al. found that use of TXA on the day of surgery in TKA was associated with significantly diminished odds of PJI in the first 3 months. Some 46% received TXA on the day of surgery, and 0.13% developed PJI within 3 months. After adjusting for individual and hospital-related covariates, TXA administration was related to significantly lower odds of PJI within 3 months of surgery. They concluded that TXA might play a significant role in decreasing PJI after TKA [36]. Figure 3 summarizes the role of TXA for the prevention of PJI following TKA. Table 2 summarizes patient-related and surgical risk factors for PJI after primary TKA.

### 5.2. Prognostic Nutritional Index as a Predictor of Postoperative PJI

According to Hanada et al., individuals with malnutrition have an elevated risk of postoperative adverse events after TKA. In addition, serum albumin and total lymphocyte count are deemed preoperative nutritional evaluation parameters. The prognostic nutritional index (PNI) is estimated by combining serum albumin and total lymphocyte counts. The objective of this investigation was to detect risk factors for postoperative adverse events after TKA, including preoperative nutritional evaluation, and to evaluate preoperative PNI as a predictor of postoperative adverse events [37]. A total of 160 individuals (234 knees) undergoing primary TKA were analyzed. The serum albumin (g/dL) and total lymphocyte count (/mm^3^) were studied within 90 days prior to TKA; then, the PNI was estimated. Postoperative aseptic wound complications were studied, such as skin erosion and dehiscence within 14 days and PJI following TKA. PJIs occurred in 14 (6%) knees. Postoperative aseptic wound complications within 14 days were significant risk factors for PJI. No significant dissimilarities in individual demographics, such as age, gender, BMI, or comorbidities were observed between patients with and without PJI except for the percentage of aseptic surgical wound complications. In addition, postoperative aseptic wound problems were affected by elevated BMI and low PNI. They concluded that pre-operative nutritional status, as shown by PNI and BMI, was related to postoperative wound complications within 14 days. PJI following TKA was related to early postoperative aseptic wound complications [37].

### 5.3. BMI Is a Superior Predictor of PJI Risk Than Local Quantities of Adipose Tissue

According to Shearer et al., both BMI and local quantities of adiposity at the surgical area have been found to be independent risk factors for PJI after TKA [38]. They evaluated previously utilized means of determining knee adiposity and found the best measure for forecasting both surgical time span and PJI after TKA, reviewing 4745 individuals who experienced primary TKA. Individual demographic data, surgical time span, and postoperative infection status within 12 months were obtained. Preoperative weight-bearing anteroposterior (AP) and lateral X-rays were studied to detect the thickness of the prepatellar adipose tissue, the width of the tibial plateau, and the total soft tissue knee width. The knee adipose index (KAI) was estimated from the ratio of bone to total knee width. They found considerable variability in both local parameters of adiposity compared with BMI. Neither parameter of local knee adipose tissue demonstrated a substantial correlation with PJI risk. By contrast, there was a significant correlation between PJI risk and BMI > 35. The surgical time span correlated with both BMI and parameters of local adipose tissue (KAI and prepatellar fat thickness). They concluded that BMI was a superior predictor of PJI after TKA compared with local parameters of the adipose tissue of the knee joint [38].

### 5.4. American College of Surgeons National Surgical Quality Improvement Program SSI Calculator

In a study with level 3 evidence published in 2016, Wingert et al. assessed the reliability of the American College of Surgeons National Surgical Quality Improvement Program (ACS NSQIP) SSI Calculator in forecasting 30-day and 90-day postoperative infection. The minimum follow-up was 90 days [10]. Individuals who experienced a repeat surgical intervention within 90 days of the TKA and in whom at least 1 positive intraoperative culture was obtained at the time of re-intervention were deemed to have PJI. Individual-specific risk possibilities for PJI based on demographics and comorbidities were obtained from the ACS NSQIP Surgical Risk Calculator website. The ACS NSQIP Surgical Risk Calculator demonstrated only moderate reliability in forecasting 30-day PJI. For 90-day PJI, the risk calculator was also only moderate in reliability. They concluded that the ACS NSQIP Surgical Risk Calculator was only a moderate predictor of acute PJI at the 30- and 90-day intervals following primary TKA. Therefore, orthopedic surgeons should be cautious when employing this instrument as a predictive tool for PJI [10].

## 6. Discussion

A recent study confirmed a relevant agreement among European orthopedic surgeons regarding prevention of PJI after TKA, and Table 3 shows the measures recommended by these authors to decrease the risk of PJI. However, the authors also noted that there is still room for improvement [39].

Even though a number of deterrent actions during surgeries including prophylactic IV utilization of antibiotics; preoperative disinfection of the skin; and intrawound lavage with a great quantity of saline have been carried out pre-, intra- and postoperatively, the risk of infection persists [11].

Given the high individual and societal influence of PJI and revision TKA, it is encouraging to note that the percentages of PJI are decreasing over time. However, with increasing percentages of osteoarthritis and TKA worldwide, it is likely that the absolute burden of PJI will continue to grow. Therefore, there is still a need to diminish the percentages of PJI after TKA. One approach is the use of antibiotic cement, although there are still conflicting data in the literature [40,41,42,43]. An RCT showed an 87% relative risk decrease in PJI after revision TKA utilizing a dilute povidone-iodine lavage compared with saline [44]. Both procedures deserve further analysis in the context of primary TKA through large RCTs, given that they are low-cost and potentially effective interventions. Preoperative risk factors for PJI must be addressed; for example, reducing body weight [45], controlling diabetes mellitus [46], improving malnutrition [47], and stopping smoking [48]. Individuals with malnutrition have an elevated risk of postoperative infection [47,49,50], and the frequency of malnutrition in individuals experiencing TKA has been revealed to be as high as 40% [51]. Therefore, it is important to assess nutritional status among preoperative patients.

PNI has been employed to assess nutrition in individuals with heart failure [52] and who experienced gastrointestinal surgery [53]. PNI can be easily estimated with serum albumin and total lymphocyte counts and is helpful for the nutritional assessment of individuals prior to TKA [50]. In fact, PNI has been shown as a predictor of 5-year overall survival following colorectal cancer surgical procedures and postoperative delirium [54,55].

It is important to mention the Swedish nationwide plan called Prosthesis Related Infections Shall be Stopped (PRISS), which was recently reported by Thompson et al. [18]. They calculated the incidence percentage of PJI after primary TKA prior to and after PRISS. These authors observed a 2-year incidence rate of 1.45%. The incidence rate was 1.44% prior to PRISS and 1.46% after. Diagnoses were confirmed within 30 days of primary TKA in 52%, and within 90 days in 73% of cases. A similar incidence prior to and after the PRISS plan was found. In addition, the time span to diagnosis was similar throughout both time intervals [19]. The likelihood of PJI after primary TKA can also be reduced by decreasing the patient’s weight, which will likewise minimize the risk of implant failure [17].

Kirschbaum et al. observed that the likelihood of survival of primary TKA is substantially diminished with each consecutive revision and also that PJI is the principal source of multiple revisions [56]. Muwanis et al. found that dilute povidone-iodine (Betadine, Avrio Health L.P, Stamford, CT, USA) compared with normal saline irrigation is an economical and simple technique to reduce PJI and more specifically SSI in TKA [57]. According to Buchalter et al., in spite of the utilization of topical irrigation solutions and addition of local antimicrobial agents, the use of a non-cephalosporin perioperative antibiotic (either vancomycin or clindamycin) is related to a higher risk of TKA PJI compared with cefazolin [58]. An increased frequency of PJI in individuals experiencing mobilization under anesthesia (MUA) was reported by Parkulo et al. [59].

Kurz et al. determined that intra-articular injections of hyaluronic acid or corticosteroid given within the 4-month period before TKA were not related to a high PJI risk within the elderly Medicare patient population [60]. According to Avila et al., individuals receiving intra-articular injections should wait at least 3 months prior to experiencing TKA to lessen infection risk [61]. Yang et al. reported that intra-articular injections of corticosteroid or hyaluronic acid prior to TKA increased the risk of postoperative infection. Injections given more than 3 months prior to TKA did not substantially augment the risk of infection [62].

Colonoscopy has been related to an increased PJI risk in TKA recipients [63]. The utilization of the Surgical Helmet Systems was related to a lower percentage of PJI following primary TKA than with conventional surgical gowning [64]. According to Blanchard et al., individuals with preoperative urinary tract infection within 1 week of TKA have an increased risk of postoperative PJI. Moreover, antibiotics do not seem to lessen the risk [65]. Individuals with a higher number of reported allergies could be at a higher risk of PJI after TKA [66].

The main limitation of this article is that the selection of studies that were finally analyzed was subjective, i.e., those that we considered most directly related to the title of the article were chosen. Therefore, it is possible that some important articles were not included. This article is not a systematic review of the literature, but a narrative review of the articles we found most interesting.

## 7. Conclusions

PJI is a serious adverse event following primary TKA. It has been found that 1.41% of patients experience revision TKA for PJI. The reported cumulative frequency for PJI is 0.51% at 1 year, 1.12% at 5 years, 1.49% at 10 years, and 1.65% at 15 years. The infection frequency within 2 years is 1.55%, and the frequency between 2 and up to 10 years is 0.46%.

Male gender, younger age, type II diabetes, posttraumatic arthritis, patellar resurfacing, discharge to a nursing home, obesity class II, hypertension, prior steroid therapy, rheumatoid arthritis, procedure type (bilateral), LOS longer than 35 days, prolonged operative time, current tobacco use, intra-articular injections before TKA, previous knee infections, previous multi-ligament knee surgery and utilization of blood transfusions have all been related to an increased risk of PJI. Other independent risk factors for PJI (in diminishing order of importance) are congestive heart failure, chronic pulmonary illness, pre-operative anemia, depression, renal illness, pulmonary circulation disorders, psychoses, metastatic tumor, peripheral vascular illness, and valvular illness.

Greater glucose variability in the postoperative phase has also been related to higher percentages of PJI, with hypoalbuminemia a reliable predictor. Preoperative nutritional status, as shown by PNI and BMI, is related to postoperative wound complications within 14 days. PJI following TKA has been related to early postoperative aseptic wound complications. Pre-operatively intravenously administered tranexamic acid decreases the risk of delayed PJI.

The likelihood of PJI after primary TKA can be reduced by decreasing the patient’s weight, which will also minimize the risk of implant failure. The likelihood of survival of primary TKA is substantially diminished with each consecutive revision, and PJI is the principal source of multiple revisions. Dilute povidone-iodine compared with normal saline irrigation is an economical and easy technique to reduce any PJI and more especially SSI. The utilization of a non-cephalosporin perioperative antibiotic (either vancomycin or clindamycin) is related to a higher risk of TKA PJI compared with cefazolin. An increased frequency of PJI in individuals experiencing MUA has been reported.

Intra-articular injections of hyaluronic acid or corticosteroid given within the 4-month period before TKA are not associated with higher PJI risk within the elderly Medicare patient population. Individuals receiving intra-articular injections should wait at least 3 months prior to undergoing TKA to mitigate the infection risk. Intra-articular injections of corticosteroid or hyaluronic acid prior to TKA augment the risk of postoperative infection. Injections given more than 3 months prior to TKA do not significantly augment the risk of infection.

Colonoscopy has been associated with an increased PJI risk in TKA recipients. The utilization of the Surgical Helmet Systems has been associated with an inferior percentage of PJI following primary TKA than conventional surgical gowning. Individuals with pre-operative urinary tract infection within 1 week of TKA have an increased risk of postoperative PJI. Moreover, antibiotics do not appear to mitigate this risk. Individuals with a higher number of reported allergies might be at increased risk of PJI after TKA.

The main limitation of this article is that the selection of articles that were ultimately analyzed was subjective, i.e., those that we considered most directly related to the title of the article. Therefore, it is possible that some important articles were not included in the end. This article is not a systematic review of the literature but a narrative review of the articles we found most relevant.

Of all the aforementioned risk factors, some are modifiable, and others are not. To minimize the risk of PJI, modifiable factors must be reversed or controlled (Table 4). The risk of PJI after TKA has diminished in small but uniform amounts over the past 15 years. The majority of PJIs are diagnosed within the first 2 years postoperatively, although a slight percentage continues to happen after a decade. The frequency of PJI has diminished barely over the past 15 years, it endures as one of the most disturbing adverse events of TKA, and continuous research to reduce its occurrence is needed. It is essential to be conscious of the risk factors for PJI after primary TKA, as discussed in this article, and to manage them as well as possible before surgery. It is also important for patients undergoing TKA to know to some extent their risk of developing PJI.

## Figures and Tables

**Figure 1 jcm-11-06128-f001:**
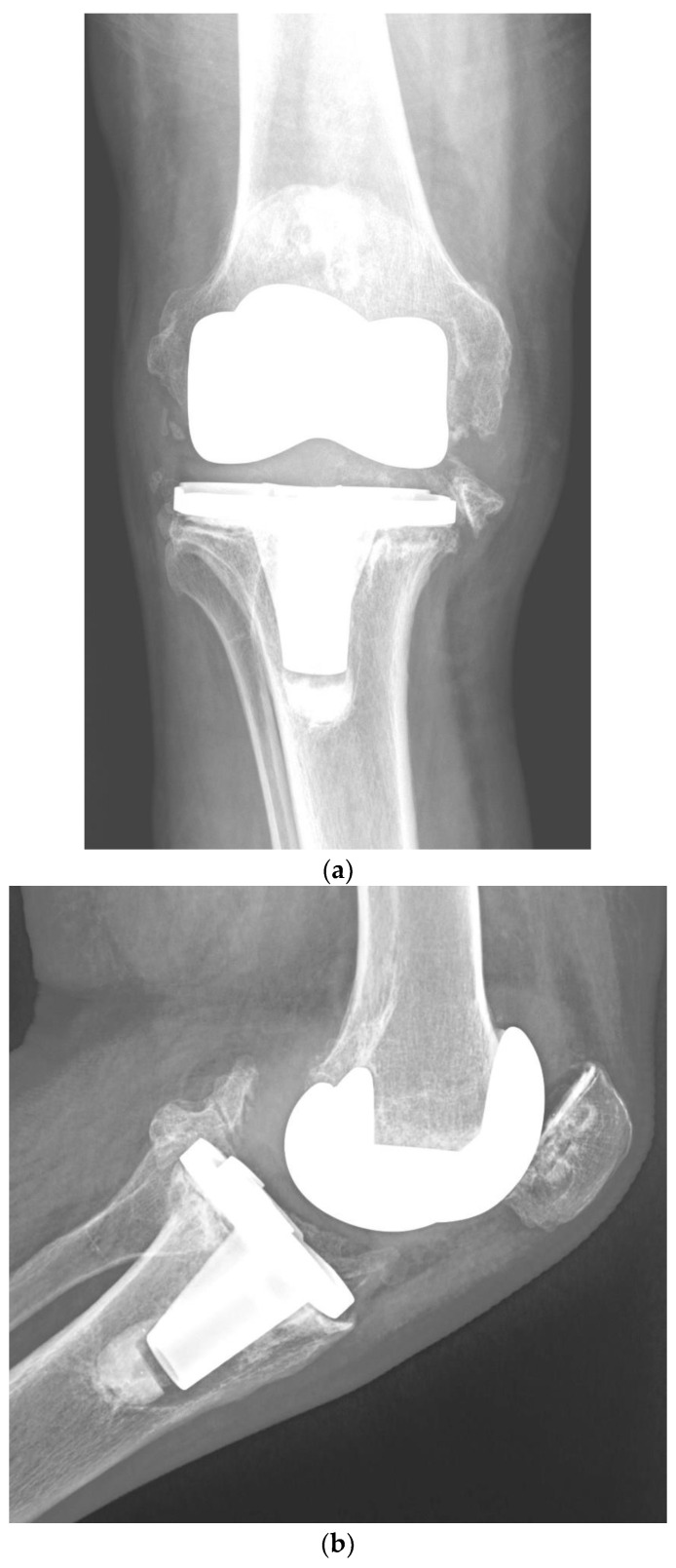
(**a**–**d**). Periprosthetic joint infection of a primary total knee arthroplasty (TKA) that was resolved by a two-stage revision TKA: (**a**) preoperative anteroposterior (AP) radiograph; (**b**) preoperative lateral image; (**c**) postoperative AP radiograph showing the prosthesis implanted in the second-stage revision (rotational hinge design); (**d**) lateral image of the aforementioned prosthesis.

**Figure 2 jcm-11-06128-f002:**
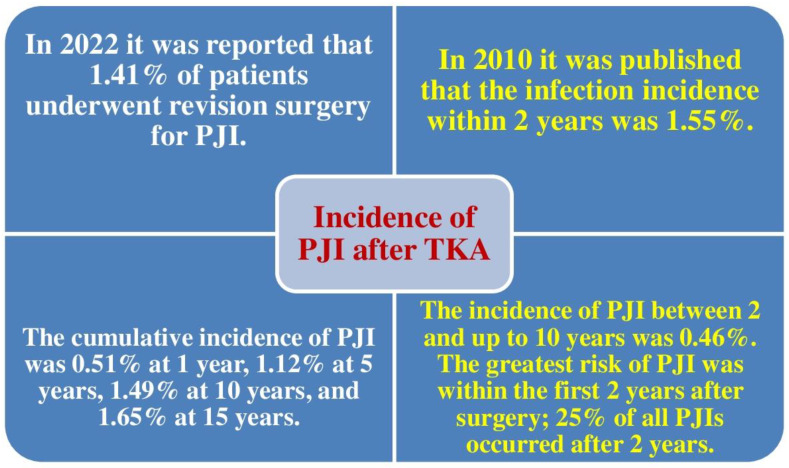
Rates of periprosthetic joint infection (PJI) after primary total knee arthroplasty (TKA) published in 2010 [20] and 2022 [11].

**Figure 3 jcm-11-06128-f003:**
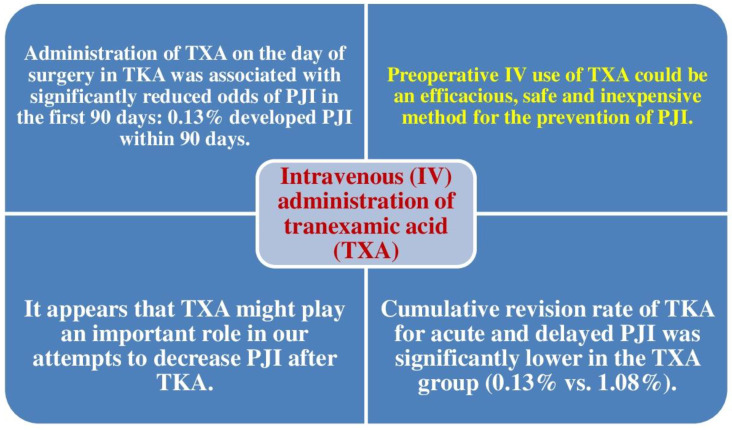
Intravenous administration of tranexamic acid appears to play an important role in the prevention of periprosthetic joint infection (PJI) after primary total knee arthroplasty (TKA) [35,36].

**Table 1 jcm-11-06128-t001:** MSIS definition of PJI [7].

PJI Exists When	
**1**	There is a sinus tract communicating with the implant; or
**2**	A bacterium is isolated by culture from 2 or more separate tissue or fluid samples attained from the affected knee; or
**3**	When 4 of the following 6 criteria exist: Raised serum erythrocyte sedimentation rate and serum C-reactive protein concentration Raised synovial white blood cell count Raised synovial polymorphonuclear percentage Existence of purulence in the affected joint Isolation of a microorganism in one culture of periprosthetic tissue or fluid Greater than 5 neutrophils per high-power field in 5 high-power fields noticed from histologic analysis of periprosthetic tissue at ×400 magnification

**Table 2 jcm-11-06128-t002:** Patient-related and surgical risk factors for PJI after primary TKA, as well as risk factors in the postoperative period.

	Risk Factors
Patient-related risk factors	Male gender Younger age Type II diabetes Obesity class II Hypertension Hypoalbuminemia Poor preoperative nutritional status Rheumatoid arthritis Post-traumatic osteoarthritis Intraarticular injections before TKA Previous multi-ligament knee surgery Previous septic arthritis Previous steroid therapy Current tobacco use Congestive heart failure Chronic pulmonary disease Preoperative anemia Depression Renal illness Pulmonary circulation disorders Psychoses Metastatic tumor Peripheral vascular illness Valvular illness
Surgical risk factors	Prolonged operative time Procedure type (bilateral) Patellar resurfacing
Risk factors in the postoperative period	Use of blood transfusion Higher glucose variability Length of stay over 35 days Discharge to convalescent care

**Table 3 jcm-11-06128-t003:** Measures recommended to minimize the risk of periprosthetic joint infection (PJI) following TKA [38].

Measures
Changeable Risk Factors Should Be Optimized before TKA
Patient education should involve skin cleaning methods with a remnant antiseptic solution
Alcoholic chlorhexidine provides better protection than alcoholic povidone-iodine against PJI
Alcohol-based solutions should be utilized in surgical hand preparation
A standardized method to the utilize of antiseptics should be in place, with special attention to the incision area
Antibiotic prophylaxis should be given before surgery and not routinely prolonged
Traffic and number of personnel in the operating room should be maintained to a minimum
Tranexamic acid and hemostatic drug utilization should be optimized to diminish the need for a surgical drain
Structured monitoring and reporting protocols for PJI should be in place
Specific instructions for PJI should be created and executed; these should be tailored to individual patient risk factors
Instructions based on level 1 or 2 of evidence should be deemed compulsory
Infections that appear 30 days after surgery can still be deemed to be PJI

**Table 4 jcm-11-06128-t004:** Main modifiable risk factors of periprosthetic joint infection (PJI) before surgery.

Risk Factor	Control Needed
Hyperglycemia	Control preoperatively
Obesity	Try to control
Hypertension	Control preoperatively
Previous intra-articular injections	Avoid 6 months before
Hypoalbuminemia	Unknown if control decreases risk
Tobacco use	Cessation of smoking at least 1 month before
Previous infection	Wait at least 3 months after infection is resolved
Nutritional status	Unknown if control decreases risk
Preoperative anemia	Correct preoperatively
Steroid therapy	Avoid for 1 month before, if possible

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
