# Peer review of "Risk Factors for Periprosthetic Joint Infection after Primary Total Knee Arthroplasty"

_jcm, 2022, doi:10.3390/jcm11206128_

Round 1

Reviewer 1 Report

The article is a narrative review of risk factors for PJI of TKA.

I think the state of knowledge on the topic and the presence of solid studies with quantitative approach in the Literature, makes a narrative review unnecessary. In my opinion, one of the purposes of narrative reviews is to provide an overview of a topic by exposing its main points, in order to facilitate the reader's possible deeper investigation of specific aspects later. In this case, I think it is relatively simple to find the information on the topic contained in this article. In addition, I think the article is nonlinear in structure and methodologically unclear.

Consequently, although I commend the authors for the extensive compilative work certainly worthy of mention.

Author Response

REVIEWER-1 (REVISION-1 submitted 6 October 2022)

The article is a narrative review of risk factors for PJI of TKA.

I think the state of knowledge on the topic and the presence of solid studies with quantitative approach in the Literature, makes a narrative review unnecessary. In my opinion, one of the purposes of narrative reviews is to provide an overview of a topic by exposing its main points, in order to facilitate the reader's possible deeper investigation of specific aspects later. In this case, I think it is relatively simple to find the information on the topic contained in this article. In addition, I think the article is nonlinear in structure and methodologically unclear.

Consequently, although I commend the authors for the extensive compilative work certainly worthy of mention.

AUTHORS: We believe that the structure is linear (with the sections of the article shown below) and the methodology is clear (narrative review of literature).  Besides, we have added the following paragraph (IN BLUE)

In other words, the purpose of this narrative review is to provide an overview of the risk factors for PJI after primary TKA. To this end, we have outlined the most important points that may facilitate the reader to further investigate specific aspects of the topic in more depth. 

STRUCTURE

Introduction

Incidence of PJI after TKA

Predictors of periprosthetic joint infection after primary TKA

Patient-related risks factors

*Male sex, procedure type (bilateral), length of stay (LOS) over 35 days, and usage of transfusions have been demonstrated to be risk factors for post-operative PJI

*Male gender, younger age, type II diabetes, post-traumatic osteoarthritis, patellar resurfacing, and discharge to nursing home were associated with augmented risk of PJI

*Previous septic arthritis has been demonstrated to be a risk factor for post-operative PJI

*Smoking is related to higher percentages of PJI

*Hypoalbuminaemia and obesity class II are dependable predictors of PJI 

*Intra-articular injections prior to TKA are related to a higher risk of PJI

*Greater glucose variability in the post-operative period is related to augmented percentages of PJI

*Reduction of patient´s weight diminishes the probability of PJI and minimize implant failure.

Surgical risk factors

*Prolonged surgical time correlates with augmented infection percentage

*Unilateral versus bilateral TKA

*Patellar resurfacing

Risk factors in the post-operative phase

*Use of blood transfusion

*Higher glucose variability

*Length of stay (LOS) over 35 days

*Discharge to convalescent care

Other topics of interest related to the risk of PJI after TKA

Tranexamic acid diminishes the risk of revision TKA for acute and late PJI

Prognostic nutritional index as a predictor of post-operative PJI

BMI is a superior predictor of PJI risk than local quantities of adipose tissue

American College of Surgeons National Surgical Quality Improvement Program (ACS NSQIP) Surgical Site Infection Calculator

Discussion

Conclusions

References

AUTHORS: Regarding the methodology, we think that it is clear now with the following paragraph:

METHODOLOGY

A PubMed (MEDLINE), Cochrane Library, Web of Science, and Scopus search of reports on PJI in TKA was examined. The key words utilized were “PJI TKA risk factors”. The principal inclusion criteria were that the reports were focused on the risk factors of PJI in TKA. Studies not focused on such risk factors were excluded. The searches were from the creation of the search engines until 30 September 2022. In other words, from the 13304 articles (10300 in the Web of Science, 2860 in Scopus, 136 in PubMed, 8 in The Cochrane Library), we chose those that seemed most directly related to the title of this article.

Reviewer 2 Report

1.      The article title's uppercase and lowercase should be changed according to MDPI format.

2.      Please include all of the author’s emails after affiliation with name initials, except for the corresponding author based on MDPI format.

3.      Please conclude your abstract with a "take-home" message.

4.      Keywords should be reorganized alphabetically.

5.      Please use lowercase font for each term following MDPI format.

6.      It is unclear whether the author's something new in this review. According to evaluation, several published review by other researchers in the past adequately explain the issues you made in the present paper. Please be careful to highlight in the introduction section anything really innovative in this review.

7.      The end of a paragraph in the introduction section should explain the objective of the present article, the present form was not.

8.      Joint infection after primary surgery would be reduced from patient aspect, such as reducing their weight that reducing the probability of periprosthetic joint infection and minimize implant failure. The introduction and/or discussion part of an article should contain this crucial topic, according to the authors. In addition, to reinforce this explanation, the recommended reference should be cite as follows: Ammarullah, M. I.; Santoso, G.; Sugiharto, S.; Supriyono, T.; Kurdi, O.; Tauviqirrahman, M.; Winarni, T. I.; Jamari, J. Tresca Stress Study of CoCrMo-on-CoCrMo Bearings Based on Body Mass Index Using 2D Computational Model. Jurnal Tribologi 2022, 33, 31–8. https://jurnaltribologi.mytribos.org/v33/JT-33-31-38.pdf

9.      In line 74, the authors need to add resource from main database apart of PubMed, there are Web of Science and Scopus. It is mandatory changed that needs to done by the authors after revision.

10.   What is the limitation of the present work? Please include it before the conclusion section.

11.   In the conclusion section, further research must be discussed.

12.   The reference should be enriched with literature from the last five years. Literature published by MDPI is strongly recommended.

13.   The authors occasionally created paragraphs in the entire document that were just one or two phrases long, which made the explanation difficult to understand. To make their explanation into a longer, more thorough paragraph, the authors should expand it. It is advised to use at least three sentences in a paragraph, with one serving as the primary sentence and the others as supporting phrases.

14.   Because of grammatical faults and linguistic style, the authors must proofread the document. MDPI English editing service would be a solution.

15.   Please review and confirm that the writers followed the MDPI format exactly, edit the current form, and recheck in addition to the other issues that have been mentioned.

Author Response

REVIEWER-2 (REVISION-1 submitted 6 October 2022)

  1. The article title's uppercase and lowercase should be changed according to MDPI format.

AUTHORS: It has been changed.

  1. Please include all of the author’s emails after affiliation with name initials, except for the corresponding author based on MDPI format.

AUTHORS: It has been done.

  1. Please conclude your abstract with a "take-home" message.

AUTHORS: We have included the following sentences (IN RED):

Knowing the risk factors for PJI after TKA, especially those that are avoidable or controllable, is critical to minimising (ideally avoiding) this terrible complication. These risk factors are outlined in this article.

  1. Keywords should be reorganized alphabetically.

AUTHORS: It has been done.

  1. Please use lowercase font for each term following MDPI format.

AUTHORS: It has been done.

  1. It is unclear whether the author's something new in this review. According to evaluation, several published review by other researchers in the past adequately explain the issues you made in the present paper. Please be careful to highlight in the introduction section anything really innovative in this review.

AUTHORS: We have included the following paragraph (IN RED):

This article seeks to understand all the risk factors for PJI after primary TKA in order to take them into account and try to control or avoid those that may be controllable or avoidable. This is the most remarkable and novel aspect of this article, being a compilation of information of great importance to fight against this terrible complication.

  1. The end of a paragraph in the introduction section should explain the objective of the present article, the present form was not.

AUTHORS: We have included the following sentences (IN BLUE) because it is the same response that we gave to Reviewer-1.

The purpose of this narrative review is to provide an overview of the risk factors for PJI after primary TKA. To this end, we have outlined the most important points that may facilitate the reader to further investigate specific aspects of the topic in more depth. 

  1. Joint infection after primary surgery would be reduced from patient aspect, such as reducing their weight that reducing the probability of periprosthetic joint infection and minimize implant failure. The introduction and/or discussion part of an article should contain this crucial topic, according to the authors. In addition, to reinforce this explanation, the recommended reference should be cite as follows: Ammarullah, M. I.; Santoso, G.; Sugiharto, S.; Supriyono, T.; Kurdi, O.; Tauviqirrahman, M.; Winarni, T. I.; Jamari, J. Tresca Stress Study of CoCrMo-on-CoCrMo Bearings Based on Body Mass Index Using 2D Computational Model. Jurnal Tribologi 2022, 33, 31–8. https://jurnaltribologi.mytribos.org/v33/JT-33-31-38.pdf

AUTHORS: We have included the following sentence (and the recommended new reference, wich is now number 15) both in the “Introduction”, “Patient-related risk factors” and the “Discussion”.

It is also important to note that the likelihood of PJI after primary TKA can be reduced by decreasing the patient's weight, which will also minimise the risk of implant failure.

  1. In line 74, the authors need to add resource from main database apart of PubMed, there are Web of Science and Scopus. It is mandatory changed that needs to done by the authors after revision.

AUTHORS: We have searched PubMed, Web of Science and Scopus. We have written the following sentence.

In other words, from the 13304 articles (10300 in the Web of Science, 2860 in Scopus, 136 in PubMed, 8 in The Cochrane Library), we chose those that seemed most directly related to the title of this article.

  1. What is the limitation of the present work? Please include it before the conclusion section.

AUTHORS: We have included the following paragraph:

The main limitation of this article is that the selection of articles that were finally analysed was subjective, i.e. those that we considered most directly related to the title of the article were chosen. Therefore, it is possible that some important articles were not included in the end. It should be remembered that this article is not a systematic review of the literature, but a narrative review of the articles we found more relevant.

  1. In the conclusion section, further research must be discussed.

AUTHORS: We have expanded the “Conclusions” with the following paragraphs:

The likelihood of PJI after primary TKA can be reduced by decreasing the patient's weight, which will also minimise the risk of implant failure. The likelihood of survival of primary TKA is significantly diminished with each subsequent revision, being PJI the main cause of multiple revisions. Dilute povidone-iodine compared to normal saline irrigation is an economical and simple technique to reduce any PJI and more specifically SSI. The utilization of a non-cephalosporin perioperative antibiotic (either vancomycin or clindamycin) is associated with a higher risk of TKA PJI compared to cefazolin. An increased frequency of PJI in individuals experiencing MUA has been reported.

Intra-articular injections of hyaluronic acid or corticosteroid given within the 4-month period before TKA are not associated with high PJI risk within the elderly Medicare patient population. Individuals receiving intra-articular injections should wait at least 3 months prior to experiencing TKA to mitigate infection risk. Intra-articular injections of corticosteroid or hyaluronic acid prior to TKA augment the risk of postoperative infection. Injections given more than 3 months prior to TKA do not significantly augment the risk of infection. 

Colonoscopy has been associated with an augmented PJI risk in TKA recipients. The utilization of the Surgical Helmet Systems has been associated with an inferior percentage of PJI following primary TKA than conventional surgical gowning. Individuals with pre-operative urinary tract infection (UTI) within 1 week of TKA have an augmented risk of postoperative PJI. Moreover, antibiotics do not seem to mitigate risk. Individuals with a higher number of reported allergies may be at augmented risk of PJI after TKA.

  1. The reference should be enriched with literature from the last five years. Literature published by MDPI is strongly recommended.

AUTHORS: In “Discussion” we have added the following sentences (with references) – numbers 54 to 64:

  1. Kirschbaum S, Erhart S, Perka C, Hube R, Thiele K. Failure analysis in multiple TKArevisions-Periprosthetic infections remain surgeons' nemesis. J Clin Med. 2022 Jan 13;11(2):376. doi: 10.3390/jcm11020376.PMID: 35054068

Kirschbaum et al have observed that the likelihood of survival of primary TKA is significantly diminished with each subsequent revision, and also that PJI is the main cause of multiple revisions.

  1. Muwanis M, Barimani B, Luo L, Wang CK, Dimentberg R, Albers A. Povidone-iodine irrigation reduces infection after total hip and knee arthroplasty. Arch Orthop Trauma Surg. 2022 Apr 30. doi: 10.1007/s00402-022-04451-z. Online ahead of print.PMID: 35488919

Muwanis et al have encountered that dilute povidone-iodine (Betadine, Avrio Health L.P, Stamford, CT) compared to normal saline irrigation is an economical and simple technique to reduce any PJI and more specifically SSI inTKA.

  1. Buchalter DB, Nduaguba A, Teo GM, Kugelman D, Aggarwal VK, Long WJ. Cefazolin remains the linchpin for preventing acute periprosthetic joint infection following primary total knee arthroplasty. Bone Jt Open. 2022 Jan;3(1):35-41. doi: 10.1302/2633-1462.31.BJO-2021-0051.R1.PMID: 35014563

According to Buchalter et al, in spite of the utilization of topical irrigant solutions and addition of local antimicrobial agents, the utilization of a non-cephalosporin perioperative antibiotic (either vancomycin or clindamycin) is associated with a higher risk of TKA PJI compared to cefazolin.

  1. Parkulo TD, Likine E, Ong KL, Watson H, Smith LS, Malkani AL. Manipulation following primary total knee arthroplasty is associated with increased rates of infection and revision. J Arthroplasty. 2022 Sep 30:S0883-5403(22)00897-X. doi: 10.1016/j.arth.2022.09.027. Online ahead of print.PMID: 36191695

An increased frequency of PJI in individuals experiencing mobilization under anaesthesia (MUA) has been reported by Parkulo et al.

  1. Kurtz SM, Mont MA, Chen AF, Valle CD, Sodhi N, Lau E, Ong KL. Intra-articular corticosteroid or hyaluronic acid injections are not associated with periprosthetic joint infection risk following total knee arthroplasty. J Knee Surg. 2022 Jul;35(9):983-996. doi: 10.1055/s-0040-1721128. Epub 2021 Jan 3.PMID: 33389729

Kurz et al encountered that intra-articular injections of hyaluronic acid or corticosteroid given within the 4-month period beforeTKA were not associated with high PJI risk within the elderly Medicare patient population.

  1. Avila A, Acuña AJ, Do MT, Samuel LT, Kamath AF. Intra-articular injection receipt within 3 months prior to primary total knee arthroplasty is associated with increased periprosthetic joint infection risk. Knee Surg Sports Traumatol Arthrosc. 2022 Mar 24. doi: 10.1007/s00167-022-06942-3. Online ahead of print.PMID: 35325263

According to Avila et al individuals receiving intra-articular injections should wait at least 3 months prior to experiencing TKA to mitigate infection risk.

  1. Yang X, Li L, Ren X, Nie L. Do preoperative intra-articular injections of corticosteroids or hyaluronic acid increase the risk of infection after total knee arthroplasty? A meta-analysis. Bone Joint Res. 2022 Mar;11(3):171-179. doi: 10.1302/2046-3758.113.BJR-2021-0350.R1.PMID: 35311571

Yang et al have reported that intra-articular injections of corticosteroid or hyaluronic acid prior to TKA augment the risk of postoperative infection. Injections given more than 3 months prior to TKA did not significantly augment the risk of infection. 

  1. Shin KH, Han SB, Song JE. Risk of periprosthetic joint infection in patients with total knee arthroplasty undergoing colonoscopy: a nationwide propensity score matched study. J Arthroplasty. 2022 Jan;37(1):49-56. doi: 10.1016/j.arth.2021.09.015. Epub 2021 Sep 27.PMID: 34592355

Colonoscopy has been associated with an augmented PJI risk in TKA recipients.

  1. Rahardja R, Morris AJ, Hooper GJ, Grae N, Frampton CM, Young SW. Surgical helmet systems are associated with a lower rate of prosthetic joint infection after total knee arthroplasty: combined results From the New Zealand Joint Registry and Surgical Site Infection Improvement Programme. J Arthroplasty. 2022 May;37(5):930-935.e1. doi: 10.1016/j.arth.2022.01.046. Epub 2022 Jan 25.PMID: 35091034

The utilization of the Surgical Helmet Systems was associated with an inferior percentage of PJI following primary TKA than conventional surgical gowning.

  1. Blanchard NP, Browne JA, Werner BC. The timing of preoperative urinary tract infection influences the risk of prosthetic joint infection following primary total hip and knee arthroplasty. J Arthroplasty. 2022 May 19:S0883-5403(22)00586-1. doi: 10.1016/j.arth.2022.05.034. Online ahead of print.PMID: 35598757

According to Blanchard et al, individuals with pre-operative urinary tract infection (UTI) within 1 week of TKA have an augmented risk of postoperative PJI. Moreover, antibiotics do not seem to mitigate risk.

  1. Fisher ND, Bi AS, Singh V, Sicat CS, Schwarzkopf R, Aggarwal VK, Rozell JC. Are patient-reported drug allergies associated with prosthetic joint infections and functional outcomes following total hip and knee arthroplasty? J Arthroplasty. 2022 Jan;37(1):26-30. doi: 10.1016/j.arth.2021.09.008. Epub 2021 Sep 20.PMID: 34547427

Individuals with a higher number of reported allergies may be at augmented risk of PJI after TKA.

  1. The authors occasionally created paragraphs in the entire document that were just one or two phrases long, which made the explanation difficult to understand. To make their explanation into a longer, more thorough paragraph, the authors should expand it. It is advised to use at least three sentences in a paragraph, with one serving as the primary sentence and the others as supporting phrases.

AUTHORS: This has been amended the best we could.

  1. Because of grammatical faults and linguistic style, the authors must proofread the document. MDPI English editing service would be a solution.

AUTHORS: The paper was revised by a native English speaker.

  1. Please review and confirm that the writers followed the MDPI format exactly, edit the current form, and recheck in addition to the other issues that have been mentioned.

AUTHORS: We think we have followed the MDPI format exactly, edited the current form, and rechecked in addition to the other issues that have been mentioned.

Round 2

Reviewer 1 Report

I think the responses to the reviewers have greatly improved the article. However, I strongly encourage the authors to use and describe in the text the most recent definitions and criteria for PJI, as in the references below: 

Parvizi J, Tan TL, Goswami K, et al. The 2018 Definition of Periprosthetic Hip and Knee Infection: An Evidence-Based and Validated Criteria. J Arthroplasty. 2018;33(5):1309-1314.e2. doi:10.1016/j.arth.2018.02.078

McNally M, Sousa R, Wouthuyzen-Bakker M, et al. The EBJIS definition of periprosthetic joint infection. Bone Joint J. 2021;103-B(1):18-25. doi:10.1302/0301-620X.103B1.BJJ-2020-1381.R1

Author Response

I think the responses to the reviewers have greatly improved the article. However, I strongly encourage the authors to use and describe in the text the most recent definitions and criteria for PJI, as in the references below: 

Parvizi J, Tan TL, Goswami K, et al. The 2018 Definition of Periprosthetic Hip and Knee Infection: An Evidence-Based and Validated Criteria. J Arthroplasty. 2018;33(5):1309-1314.e2. doi:10.1016/j.arth.2018.02.078

McNally M, Sousa R, Wouthuyzen-Bakker M, et al. The EBJIS definition of periprosthetic joint infection. Bone Joint J. 2021;103-B(1):18-25. doi:10.1302/0301-620X.103B1.BJJ-2020-1381.R1

AUTHORS: As suggested by the Reviewer, we have described in two paragraphs of the INTRODUCTION (IN BLUE) the most recent definitions of PJI according to Parvizi et al, and McNally et al. Therefore, we have included two new references on the list of REFERENCES [8. Parvizi et al] and [9. McNally et al].

Reviewer 2 Report

Well done.

Author Response

Well done.

Extensive editing of English language and style required

AUTHORS: Thank you very much to the Reviewer for his/her positive comment.

As suggested by the Reviewer, REVISION-2 of the article has been edited by a native English interpreter.